# FiRST: Finetuning Router-Selective Transformers for Input-Adaptive Latency Reduction

## Abstract

Auto-regressive Large Language Models (LLMs) demonstrate remarkable performance across domanins such as vision and language processing. However, due to sequential processing through a stack of transformer layers, autoregressive decoding faces significant computation/latency challenges, particularly in resource-constrained environments like mobile and edge devices. Existing approaches in literature that aim to improve latency via skipping layers have two distinct flavors - 1) Early exit 2) Input-agnostic heuristics where tokens exit at pre-determined layers irrespective of input sequence. Both the above strategies have limitations - the former cannot be applied to handle KV Caching necessary for speed-ups in modern framework and the latter does not capture the variation in layer importance across tasks or more generally, across input sequences. To address both limitations, we propose FiRST, an algorithm that reduces inference latency by using layer-specific routers to select a subset of transformer layers adaptively for each input sequence - the prompt (during prefill stage) decides which layers will be skipped during decoding. FiRST preserves compatibility with KV caching enabling faster inference while being quality-aware. FiRST is model-agnostic and can be easily enabled on any pre-trained LLM. We further improve performance by incorporating LoRA adapters for fine-tuning on external datasets, enhancing task-specific accuracy while maintaining latency benefits. Our approach reveals that input adaptivity is critical - indeed, different task-specific middle layers play a crucial role in evolving hidden representations depending on task. Extensive experiments show that FiRST significantly reduces latency while retaining competitive performance (as compared to baselines), making our approach an efficient solution for LLM deployment in low-resource environments.

## 1 Introduction

Large Language Models (LLM's) have revolutionized the fields of Natural Language Processing and Computer Vision achieving incredible performance on a diverse set of benchmark tasks. However, the scale of these LLM's characterized by billions of parameters hinder their adoption in resource-constrained environments with memory, latency and compute serving as the main challenges. In this work, we focus on the latency aspect which becomes the most significant bottleneck for tasks such as machine translation, question answering, summarization particularly on devices, such as laptops and mobile phones. As mentioned in (Schuster et al., 2022), the auto-regressive nature of decoding in LLM's further pronounces the latency bottleneck. Our main interest lies in the resource-constrained on-device setting where resolving this bottleneck is of particular importance.

Transformer based LLMs have several stacks of layers (including attention and FFN layers) leading to high latency and compute requirements, making inference very slow or even infeasible in resource constrained settings. This is because of the sequential processing of tokens through all the layers for *every input sequence and task*. However, it is important to note that in the real world, there is a lot of heterogeneity in input sequences and tasks. (Schuster et al., 2022; Sun et al., 2022) noted that the generations made by LLMs can have varying levels of difficulty and certain generations can be solved with reduced compute, by exiting the transformer stack early. At the same time, it has been noted in recent works (Wendler et al., 2024) that inference forward pass proceeds in phases

through the layers of transformer based models, with different types of information being extracted or mapped at different phases (sequences of layers) for certain tasks such as translation. Motivated by these and other related works, we hypothesize that *different sequential combinations of layers are important for different input sequences and tasks*. Learning the right sequential combination of layers can help reduce inference latency and compute for on-device scenarios. However, there are several challenges. Any algorithm for determining the "right" combination of layers should minimize any quality loss, be compatible with other latency reduction strategies such as KV cache handling and batch inference, should not introduce any additional latency or compute and and be learnable with minimal compute and training overhead.

In the last few years, several promising approaches have been proposed in literature that adaptively prune layers at each decoding step. Token-level early exit proposed in (Schuster et al., 2022; Sun et al., 2022) allow tokens to exit the transformer layer stack early based on different strategies to compute the confidence or saturation level. (Elhoushi et al., 2024; Elbayad et al., 2020; Zhang et al., 2019) extended this idea to incorporate layer skipping at a token level during training. While token level early exit is a useful idea in theory, it suffers from a major limitation of incompatible KV Caching in practice (Del Corro et al., 2023). The incompatibility stems from having to recompute KV caches for preceding tokens if we have a delayed exit point for latter tokens often resulting in loss of early exit advantages. Since KV cache is crucial in significantly speeding up auto-regressive decoding, inappropriate handling of KV cache limits practical adoption.

Recently, (Liu et al., 2024; Del Corro et al., 2023) have proposed input-agnostic layer skipping at token level, that handle KV cache appropriately as well as retain the advantage of adaptive partial computation. In these solutions, tokens exit at pre-determined layers irrespective of the input sequence, and for all sequences in a batch, tokens at the same position in a sequence exit at the same layer. Furthermore, tokens at latter parts of the sequence are constrained to exit earlier than the previous tokens to ensure that there is no redundant KV cache re-computation. These solutions are heuristic based and impose hard rules and constraints irrespective of input sequences, which can lead to drop in output quality. Others (Jaiswal et al., 2024) have proposed circumventing the KV cache issue entirely by skipping only FFN layers, but such a strategy cannot reduce redundancy in transformer layer computations. Moreover, they propose an input adaptive skipping heuristic based on cosine similarity of outputs: if two adjacent layers have a similarity greater than a threshold, then all subsequent layers except the last few are skipped. However, such a strategy does not take into account that several middle layers are crucial (see (Liu et al., 2024)) and furthermore, final prediction capability of full model is not taken into account while deciding which layers to skip.

Our goal is to design an input-adaptive **learnable layer selection strategy** with quality aware latency gains that is also able to handle the KV Cache appropriately. Ideally, for every input sequence and task, we want to predict the optimal (sequential) combination of layers at inference time, such that quality loss is minimum and the latency gains are as high as possible. We want to do this with expending very little compute/additional training, with no or minimal additional latency (for inference) and handle KV cache appropriately. Since there are exponential number of possible layer sequences, this seems like a hard goal computationally and otherwise - however, we have addressed this layer selection challenge partially in this work. We propose an approach for learning and predicting the layer selections based on the input sequence and task, via training **routers**. Based on the output of each layer for a sequence, a router will decide whether or not to skip the subsequent layer in the transformer architecture. Since the decision is at a sequence level, KV cache issues do not arise, as all tokens in a sequence would pass through the same set of layers. Moreover, we further generalize our approach to handle batch inference by making the layer selection unified for all sequences in a batch. Finally, we fine-tune the model combined with trained routers using LoRA adapters to improve the quality significantly while retaining the latency gains. As an added bonus, LoRA finetuning smoothens the layer skipping [1], reduces the amount of performance degradation and further highlights the varied importance of layers based on input sequence.

We summarize our contributions below:

1. We propose a training and inference algorithm FIRST that incorporates layer-specific routers for selecting layers in an input-adaptive manner. The layer selection is uniform for all tokens in a

---

[1]Depending on the task, after LoRA finetuning, task-specific sequences skip a certain set of middle layers significantly more than other middle layers

sequence, thus handling KV caching and batch decoding without introducing additional compute and latency. This can be applied on top of any pre-trained model.

2. We further incorporate LoRA adapters on top of the router-based layer selection for finetuning on external dataset - the goal is to improve the quality of performance of the router-augmented model on the task-specific data while retaining latency gains. This further smoothens the layer selection and an important insight emerges: certain layers including several middle layers (sequence-dependent) are significantly more important for evolving the hidden representation.

3. Finally, we demonstrate extensive experimental evidence on multiple datasets on Machine Translation and Summarization tasks answering the efficacy of FIRST in achieving good latency gains while retaining comparable quality of performance.

## 2 RELATED WORK

**Early Exit:** Several works have been proposed in the early exit theme (Zhu, 2021; Zhou et al., 2020; Xin et al., 2020; Liu et al., 2020; Li et al., 2020; Hou et al., 2020; Schuster et al., 2022) where adaptive compute is used for different parts of the token sequence. While these approaches have been popular for encoder-only models which processes the entire sequence as a whole, they have faced challenges in generation tasks. The main limitation of these set of techniques are their inability to handle KV caching appropriately which is crucial for multi-fold speed-ups in current LLM architectures. We emphasize that in our work, we assign varying compute to sequences in different batches but within the same sequence, we assign the same compute to every token.

**Input Agnostic Heuristics:** In Skip Decoding (Del Corro et al., 2023), initial tokens pass through more layers than later ones, contradicting the observation that later tokens are harder to decode (Liu et al., 2024). Additionally, Skip Decoding skips several bottom layers for most tokens, causing undesirable sub-network imbalance. To address this, Unified Layer Skipping (Liu et al., 2024) proposes a discrete skipping strategy that is uniform for all tokens in a sequence. Based on a latency budget, retained layer ids are passed through by all tokens, ensuring KV Cache handling and retaining key layers. However, the limitation of this approach is that skipping is independent of the input sequence. In contrast, early exit strategies adapt layer skipping to the input sequence, offering more flexibility. In (Fan et al., 2019), a method akin to dropout randomly skips layers during training, but this leads to performance decline during the pre-fill stage. FFN-SkipLLM (Jaiswal et al., 2024) constrains skipping to FFN layers to avoid KV Cache issues but fails to fully exploit redundancy as discussed already.

**Model Compression and Quantization Aware Training:** Orthogonal approaches to explore the latency/memory-performance trade-off in Large Language Models aim to build smaller models that approximate the performance of larger ones with reduced memory and latency costs. Key techniques include: 1) compressing model parameters into fewer bits (Frantar et al., 2022; Lin et al., 2024; Lee et al., 2024; Saha et al., 2023); 2) pruning the network by removing components like attention heads or neurons based on heuristics (Frantar & Alistarh, 2023; Ma et al., 2023); and 3) distilling the large model into a smaller, faster counterpart (Agarwal et al., 2023; Gu et al., 2024). For further details, we refer to the survey by (Zhu et al., 2023). A significant body of work (Dettmers et al., 2024; Liu et al., 2023b; Peri et al., 2020; Li et al., 2023) has focused on quantization-aware training to reduce memory footprints and mitigate performance loss, starting with QLoRA (Dettmers et al., 2024). In a similar vein, our work proposes fine-tuning router-augmented models to improve layer skipping and reduce performance degradation, as pre-trained models do not account for layer skipping, leading to higher degradation with vanilla skipping.

## 3 PROBLEM STATEMENT

Our goal is to exploit the heterogeneity in inputs and tasks to selectively use LLM layers in a quality aware manner for reducing inference latency and compute for on-device constraints. Ideally, we want to select an *optimal* sub-sequence of layers within a transformer architecture for a given input and task, such that the overall latency as well as expended computation are both low, while quality is comparable to the un-modified case where every input sequence passes through every layer. For ease of explanation, without loss of generality, we assume the task is same and simply consider an input sequence for describing the problem.

Let us consider an an input sequence $\mathcal{X} = \{x_1, x_2, \ldots, x_n\}$ with $n$ tokens. Let there be $m$ transformer layers in the model, where the $i^{th}$ transformer layer is represented as the function $\phi_i()$. As stated lucidly in (Wendler et al., 2024), $\mathcal{X}$ is first converted to an initial latent representation $\mathcal{H}_0 = \{H_0^1, H_0^2, \ldots, H_0^n\}$, where $H_j^0 \in \mathbb{R}^D, \forall j \in [n]$ is a look-up from a learned embedding dictionary corresponding to the $j^{th}$ token. Thereafter, every transformer layer $\phi_i()$ operates on the latent vectors $\mathcal{H}_i$ to generate the embedding for the $i^{th}$ layer as follows. For the $j^{th}$ token,

$$H_i^j = H_{i-1}^j + \phi_i(H_{i-1}^1, H_{i-1}^2, \ldots, H_{i-1}^j) \tag{1}$$

Let the (golden) output or generated sequence for an input sequence $\mathcal{X}$ that passed through all $m$ layers of the model with full computation be $\mathcal{Y}_{\mathcal{X}}^*$. Our hypothesis is that for a given input sequence (and task), there exists an optimal subsequence of functions $\mathcal{F}_{OPT}(\mathcal{X})$ out of the full sequence $\{\phi_i, i \in [m]\}$ such that the output generated by passing through this subsequence: $\mathcal{Y}_{OPT,\mathcal{X}} \approx \mathcal{Y}_{\mathcal{X}}^*$. More formally, if $Q$ is a quantitative quality measure on $\mathcal{Y}$, and $\epsilon \to 0$ is tolerance in deviation in quality from the golden output, then we hypothesize that there exists an optimal subsequence, using the minimum number of layers, $\mathcal{F}_{OPT}(\mathcal{X})$, such that:

$$Q\left(\mathcal{Y}_{OPT,\mathcal{X}}\right) \geq (1 - \epsilon)Q\left(\mathcal{Y}_{\mathcal{X}}^*\right), \forall \mathcal{X}. \tag{2}$$

The optimality above is with respect to the minimum subsequence of layers that can help achieve the above, to minimize latency while keeping quality unaffected. Note that, the optimal subsequence $\mathcal{F}_{OPT}(\mathcal{X})$ need to be obey the same autoregressive computation on previous tokens as given in Equation 1. Hence, any algorithm that determines the optimal subsequence, need to be compatible with KV cache handling, to avoid the re-computation of values for tokens preceding the current token (which is a drawback with some existing work, especially in the Early Exit literature, that choose computation or layer skipping at token level).

The potential number of subsequences for $m$ layers is $2^m$, hence a brute force approach is not only infeasible, but would also beat the purpose of such a layer selection in the first place: reducing latency and compute. In the absence of any known substructure in the behaviour of the latent layers on each input sequence, it is difficult to arrive at such an optimal solution polynomially or with low additional latency or compute, and in fact is likely to be NP-hard.

We propose to learn an approximation of the optimal subsequence of layers for any input sequence with low additional latency and minimal training.

## 4 PROPOSED SOLUTION: FIRST

Let us first understand what it entails to learn an optimal subsequence of layers for any input. Consider the full transformer sequence to be $F^* = \{\phi_1, \phi_2, \ldots, \phi_m\}$. Any optimal subsequence for an input $\mathcal{X}$: $\mathcal{F}_{OPT,\mathcal{X}}$ could be thought of as finding an optimal path through a binary tree of functions. Formally, let every level in the binary tree correspond to a transformer layer and the $0^{th}$ layer corresponds to the initial embedding look up; i.e., at depth $i \in [m]$, there would be $2^i$ nodes, each corresponding to either $\phi_i$ or $\overline{\phi_i}$, where the former denotes that a particular transformer layer is included in the optimal path whereas the latter denotes that it is not included. Each (of the $2^{i-1}$ nodes) $\phi_i$ or $\overline{\phi_i}$ has two children, corresponding to the next transformer layer: $\phi_{i+1}$ and $\overline{\phi_{i+1}}$ (See Figure 1). In such a tree structure, for example, the path $\{\phi_{i-1}, \overline{\phi_i}, \overline{\phi_{i+1}}, \phi_{i+2}\}$ indicates the subsequence of transformer layers $\{\phi_i, \phi_{i+2}\}$. For any transformer layer $\phi_i$ in this tree, let $Anc(\phi_i) = k, 0 < k < i$ denote the the lowest ancestor node where the corresponding transformer node $\phi_k$ is included in the sequence. In the above example, $Anc(\phi_{i+2}) = \phi_{i-1}$.

For any such sequence of functions $\mathcal{F}$, at level $i$, the autoregressive computations for the $j^{th}$ position corresponding to the $j^{th}$ token in the input sequence, Equation 1, would now be modified as follows.

$$H_i^j = \begin{cases} H_k^j & \text{if } \phi_i \notin \mathcal{F}, \ k = Anc(\phi_i) \\ H_k^j + \phi_i(H_k^1, H_k^2, \ldots, H_k^j), & \text{if } \phi_i \in \mathcal{F}, \ k = Anc(\phi_i) \end{cases} \tag{3}$$

Our problem translates to navigating this binary tree to find the optimal path $\mathcal{F}_{OPT}$ for any given input sequence and task. Since there are $2^m$ paths in this tree, we propose to approximate the

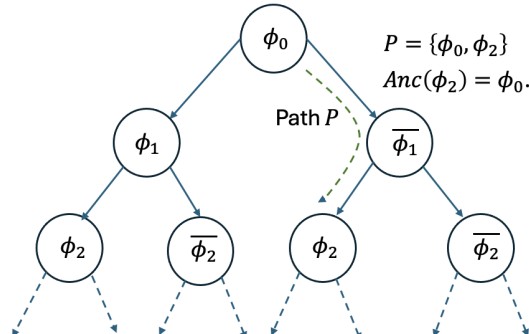

Figure 1: Binary Tree representation of the subsequence of layer selection.

optimal choice by making this decision in a greedy fashion at each node. Formally, we propose to add a (lightweight and fast) router $R_i$ before every transformer layer $\phi_i$ in the model, that will predict whether $\phi_i$ will be selected or not.

Our aim is to learn to predict the layer choice at each level (myopically) so that the overall path (or, sequence of functions) $\mathcal{F}(\mathcal{X}) \approx \mathcal{F}_{OPT}(\mathcal{X})$ for any given input sequence $\mathcal{X}$ (and task). We want to decide this at a sequence level, and not at a token level to maintain compatibility with the autoregressive computations and avoid re-computation of values (use KV cache efficiently). Moreover, we want to spend minimal compute and training for learning these functions $R_i$ and finally, $R_i$ functions should be lightweight and low compute so that that do not add any significant latency to the overall computation, helping realize the latency gains.

Our proposed algorithm FIRST modifies any off-shelf pre-trained transformer based model by incorporating and training a router or probability function $R_i$ before every transformer layer $\phi_i$. For a given input sequence $\mathcal{X}$, the output of the router $R_i$ is a probability score $\rho_i$ denoting the probability of selecting $\phi_i$ in the layer sequecne. We can think of layer $i$ as a modified function $\phi_i^R((X))$ such that the output is $\rho_i \cdot \phi_i(\mathcal{X}) + (1 - \rho_i) \cdot \phi_i(\mathcal{X})$. Formally, the autoregressive computation in Equation 1 would now be modified as follows.

$$H_i^j = H_{i-1}^j + \rho_i \cdot \phi_i(H_k^1, H_k^2, \ldots, H_k^j) + (1 - \rho_i) \cdot \phi_i((H_k^1, H_k^2, \ldots, H_k^j)) \qquad (4)$$

The above equation, applied recursively would be approximating the Equation 3 for the optimal $\mathcal{F}$ in a probabilistic, greedy manner. We train the functions $\rho_i$ on datasets and task, and further fine tune using LoRA adapters to make the layer selections smooth and improve the output quality. We explain the framework for FIRST algorithm in details in the following section.

## 5 FIRST FRAMEWORK AND ALGORITHM

In this section we describe the training and inference frameworks and procedure for FIRST in details. We first describe how to train Routers to be adaptive to input sequences. Given an off-the-shelf pre-trained LLM, we propose two training phases (Figure: 2). In the first phase, we train a router for each layer that decides whether the tokens in the input sequence should skip the layer or not. In the second phase, to tackle the issue of unseen skipping during pre-training, we fine-tune the router-augmented LLM keeping router weights fixed using LoRA (low rank adapters) to ensure the model learns to perform well on the target dataset without reducing the skipping level. In other words, the LoRA fine-tuning ensures that the gap in performance with and without skipping is significantly reduced when compared to the base model. Below, we provide the details of each phase.

### 5.1 ADAPTIVE ROUTER MODULE

The adaptive router module is a single-layer neural network without bias, positioned before every layer in the model. During training of the router, all model parameters except the router weights remain frozen. For the first layer, it takes the tokenized input, and for each of the subsequent layers, it takes the output of the preceding layer as input. Mathematically speaking, for any layer $i$, given

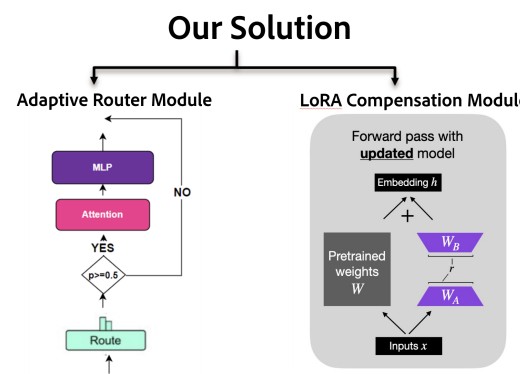

Figure 2: Layer diagram of the two training phases

a batch of $B$ tokenized inputs sequences, where each sequence has $n$ tokens and is embedded in to $\mathbb{R}^D$, the adaptive router module takes as input a $B \times n \times D$ tensor output of layer $(i-1)$ and outputs a $B \times n \times 1$ tensor. Subsequently, corresponding to each value (or, token) in the $B \times n \times 1$ tensor, we apply a sigmoid function to ensure that all entries in the tensor are in the interval $[0, 1]$. Following this, we take a mean operation at the sequence level - we take a mean of all the weights in a sequence to output a $B \times 1 \times 1$ tensor. For each sequence in the batch, the corresponding entry is the probability $\rho_i$ with which the sequence passes through the layer $i$. The input sequence skips the layer $i$ with probability $1 - \rho_i$.

In summary, after applying router $R_i$ to an input sequence at each layer $i$, a single probability value $\rho_i$ is produced, indicating whether to pass the sequence through the layer. During training, the output of a layer is modified as in Equation 4 using a skip connection, incorporating the probability $\rho_i$ (see Figure: 3). The routers are trained to encourage skipping by reducing the probabilities $\rho$ using a regularizer, to approximate the optimal subsequence for minimizing the latency. Note that the entire model is frozen except for the routers. The training task is modeled as a language modeling task, specifically next token prediction. The total loss function comprises of 3 terms namely -

- **Cross-entropy loss:** This measures the difference between the actual and predicted probability distributions, to ensure the quality of the generations. $\mathcal{L}_{\mathsf{CE}} = -\sum_{x \in \mathcal{X}} \mathcal{Y}_{\mathcal{X}}^* \log{(\hat{\mathcal{Y}})}$.
- **Regularization loss:** This adds a penalty term to the loss function so that the router parameters are not too large, to minimize the added latency and training compute, as well as minimizing overfitting to noise. $\mathcal{L}_{\mathsf{Reg}} = \sum_{i \in [m]} ||R_i||^2$, where $||R_i||^2$ denotes the $\ell_2$ norm of the router weights for the $i^{th}$ layer router, and there are $m$ layers in the model.
- **Non-skip Penalization loss:** Summation of probability values across all layers of the model architecture. It encourages the model to favor skipping at the cost of cross-entropy loss to reduce latency, with the coefficient $\alpha$, managing the extent of skipping to approximate the optimal trade-off of quality and latency. $\mathcal{L}_{\mathsf{PP}} = \sum_{i \in [m]} \rho_i$.

The total loss is a linear combination of these three terms namely $\mathcal{L} = \mathcal{L}_{\mathsf{CE}} + \lambda \cdot \mathcal{L}_{\mathsf{Reg}} + \alpha \cdot \mathcal{L}_{\mathsf{PP}}$.

## 5.2 LoRA Compensation Module

Selecting a subsequence of layers in a model to improve the latency during inference will naturally come with some performance loss - especially so since the pre-trained model was not trained to skip layers given any input sequence. To compensate for the loss in performance caused by skipping in the model, we finetune the router-augmented pre-trained model on the downstream task. We are inspired by Quantization Aware Training - QLoRA in particular - a training method which compensates for performance loss due to model compression. To finetune, we use Low Rank Adapters (LoRA) to modify the weights of the pre-trained model while keeping the router weights frozen. While using LoRA, the difference in weights for each trainable weight matrix is restricted to be a low rank matrix. During training, the router parameters are frozen while trainable LoRA adapters are added to both the FFN (Feed-Forward Network) and the attention modules of each layer of the

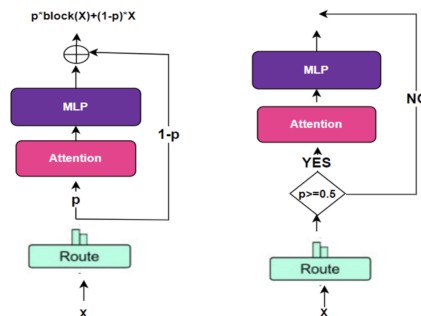

Figure 3: Skip connection used for router training. With probability $p$, the sequence is processed by the layer and with probability $1 - p$, the layer is skipped. During inference, routers make the decision of whether a sequence will skip a particular layer or pass through it.

pre-trained model. During the finetuning phase, to maintain the skipping level, we again add a similar loss component as in phase 1, namely $(\alpha/3)\mathcal{L}_{\mathsf{PP}} = (\alpha/3)\sum_{i\in[m]}\rho_i$. This is essential during the LoRA finetuning even though the router weights are fixed in this phase - this is because the finetuning mechanism alters the hidden representations of the input sequence in a manner such that the probability score for each layer is always more than $0.5$ implying that no layers are skipped. We have noticed that the Non-skip Penalization Loss coefficient $\alpha$ scaled down by a factor of 3-4 is well-suited for the finetuning process while maintaining the same skipping level as in phase 1 of the training. During training the LoRA adapters, responses are appended to the prompt to train the model to predict tokens from the response. For inference, the model weights are merged with the original weights to prevent any latency overhead.

## 5.3 Inference for FiRST

During inference, given an input sequence, the decision to skip or pass through a layer is determined by a threshold. For the input sequence, each router (corresponding to a layer) outputs a number in the interval $[0, 1]$. If this number (corresponding to the probability of not skipping the layer) is greater than or equal to 0.5, the sequence passes through the layer. On the other hand, if the output from the router is less than 0.5, the sequence skips the layer (Figure: 3). Below, we describe some salient points about the functioning of the router during inference to handle KV Cache appropriately to retain the modern latency speed-ups:

1. **Prefill phase handling:** Skipping is not allowed during prefill phase. This ensures the first token is generated correctly, which is crucial for WMT tasks, as they are highly sensitive to the correct generation of the first token in the target language. It has been observed in prior works (Liu et al., 2024) that skipping during prefill phase is detrimental to performance during inference.

2. **Fixed router decisions during decoding and handling KV Cache:** During the prefill phase, the decisions made by the routers are cached. During the decoding phase, every token adheres to the cached decision made during prefill. In other words, for a particular layer, if a router outputs a number less than 0.5 during prefill, the number is fixed for the decoding steps and therefore the same layer will be skipped by all tokens during decoding. Similarly, if the router outputs a number more than 0.5 during prefill, the same layer will be processing all tokens during decoding. Such a step ensures that for each decoding step and each layer that is not skipped, the KV cache for all previous tokens is available for that layer - this is because a fixed set of layers (decided during the prefill phase) will be skipped for all tokens during the decoding phase of inference. This approach effectively addresses the caching issues encountered in early exit strategies, ensuring consistent decisions across the decoding process.

## 6 Experiments

We conduct experiments on two benchmark tasks: Machine Translation and Text Summarization on several publicly available datasets demonstrating both robustness and scalability of FiRST.

## 6.1 DATASETS

**Machine Translation:** For translation tasks, namely English-to-Chinese and English-to-German, we employ the WMT development sets from 2017 to 2020 for training/fine-tuning following the methodology outlined in previous studies (Liu et al., 2023a; Jiao et al., 2023). Translation performance is evaluated using the test set from the WMT 2022 dataset (Kocmi et al., 2022) which was developed using recent content from diverse domains. These domains include news, social media, e-commerce, and conversational contexts. (Details in Appendix: A.1, Table: 4).

**Summarization:** We use the popular CNN-DailyMail (CNN-DM) (Hermann et al., 2015) dataset which is a large collection (over 300k) of text summarization pairs, created from CNN and Daily Mail news articles. Each datapoint in this dataset comprises of an **article** (the body of the news article with 683 words on average) and the corresponding **highlights** (article summary as written by the article author). While the training set contains more than 287k samples, we have randomly chosen 4k samples for training both routers and LoRA. During training in our framework, the number of trainable parameters is small in both phases - therefore a small subset of data points is sufficient for training. Inference is performed on the standard test set with 11,490 samples.

## 6.2 EVALUATION METRICS

**Quality-Based Metrics for Translation task:**

- **BLEU Score:** BLEU (Bilingual Evaluation Understudy) scores are used to measure the quality of translations. BLEU compares n-grams of the candidate translation to n-grams of the reference translation, providing a score between 0 and 1, with higher scores indicating better translations. In this evaluation, NLTK BLEU is employed, focusing on BLEU-1 and BLEU-2 scores.
- **COMET:** COMET (Cross-lingual Optimized Metric for Evaluation of Translation) is used to assess translation quality further. COMET evaluates translations using a model trained to correlate well with human judgments. Specifically, Unbabel/XCOMET-XL [2] is used in this evaluation. COMET provides a more nuanced assessment of translation quality by considering the intricacies of both source and target languages, beyond the n-gram matching used in BLEU.

**Quality based Metrics for Summarization Task:**

- **BERTScore:** This metric quantifies semantic similarity between texts by leveraging contextual word embeddings. BERTScore captures meaning-based similarity rather than relying on exact word matches, providing a nuanced evaluation of text generation quality.
- **ROUGE:** (Recall-Oriented Understudy for Gisting Evaluation) is a common metric - ROUGE-1 refers to overlap of unigrams between the system summary and reference summary. Similarly, ROUGE-L measures longest matching sequence of words.

Finally, for benchmarking latency, we look at the **TPOT (Time Per Output Token):** This metric evaluates the average time taken to produce each output token and is calculated for GPU to gauge overall decoding performance.

## 6.3 TRAINING AND INFERENCE SETUP

- **Training settings:** For our experiments, we use Llama-3-8B base model from Meta, which comprises of 32 layers. Training of routers and LoRA adapters is conducted on A100 80GB GPUs, with training/inference is performed in full precision to avoid performance degradation due to quantization. The training process employs our custom loss function and continues for a fixed number of epochs, terminating when the validation loss fails to improve over 4 consecutive steps. The learning rate is set between $1e^{-4}$ and $3e^{-4}$ - a cosine scheduler is used to adjust the learning rate. Gradients are accumulated after 5 steps and the regularization coefficient $\lambda$ is fixed at 0.01. For LoRA fine-tuning, we employ a rank of 8, a dropout rate of 0.1, and a scaling factor (lora_alpha) of 32. For translation, the maximum sequence length is set to 128 for router training and 256 for LoRA training. Similarly, for summarization, the maximum sequence length is set to 500 and 700 respectively. Prompts for the different tasks regarding training/inference are shown in Appendix A.2.

---

[2]https://github.com/Unbabel/COMET

- **Inference settings:** For both translation and summarization, we set the temperature to 0.8 and enable top-k sampling over 10 tokens. The maximum number of tokens to be generated is set to 80 and 200 respectively. Caching is turned on during inference.

## 6.4 BASELINES FOR COMPARISON

- **Original Model:** We compare the performance of the base model (Llama-3-8B) with and without LoRA fine-tuning as our baseline. We then train routers and the LoRA module for various values of the coefficient $\alpha$ associated with the non-skip penalization loss. This allows us to report the time per output token (TPOT) and quality at different levels of skipping, with and without LoRA. Latency speedups are reported relative to the LoRA-fine-tuned model, which natively supports key-value (KV) caching.
- **Unified Skipping:** This method relies on using a heuristic-based strategy for retaining layers at fixed intervals. We replicate their algorithm (Liu et al., 2024)and compare performance both with and without LoRA fine-tuning across various skipping percentages. We do not consider other input-agnostic heuristic-based strategy for skipping layers since Unified Skipping has empirically established itself to be the state-of-the-art.

## 6.5 DETAILED RESULTS

**Layer-wise Skipping Patterns:** First, we present some layer-wise skipping statistics across the 3 tasks that we experiment with. Note that layer-wise skipping significantly vary across tasks, indicating that the importance of each layer depends on the nature of the task and dataset. For a 15% skipping rate, we observe the following patterns. In the WMT Machine Translation task, for English-to-German translation, layers 7–9 and 21 are fully skipped, while layer 18 is partially skipped. For English-to-Chinese translation, layers 7–9, 16, and 21 are fully skipped, with partial skipping in layer 20. In the summarization task, layers 20, 22, and 23 are fully skipped, and layers 19, 21, and 26 are partially skipped. Some layers are skipped less than 5% of the time, suggesting these layers are only necessary for specific sequences, highlighting the input-dependent and task-specific nature of layer importance. Detailed layer-wise skipping statistics can be found in Appendix A.3.

Now, we present detailed analysis of our experiments on the WMT Translation and CNN Summarization datasets. We start with highlighting the salient points:

**English-to-German:**

1. We observe a performance degradation of less than 15% in COMET scores which accounts for the semantics part of the translated text. Less than 10 % and 20% degradations are observed in BLEU-1 and BLEU-2 (syntax-based metrics) respectively for approximately 15% skipping (Table: 1). The corresponding latency improvement is 10% on TPOT. (Table: 2).
2. Our approach significantly outperforms the input-agnostic layer skipping method (Liu et al., 2024), referred to as Unified Skipping. For approximately 15% skipping, our method achieves a BLEU-1 score of 38.01, compared to 28.92 for the unified skipping approach (Table: 1). In terms of COMET scores, our method attains a score of 82.14, while unified skipping achieves a score of 59.34. This demonstrates the superiority of our approach in preserving semantic integrity.

**English-to-Chinese:**

1. For approximately 15% skipping, we observe a 12% improvement in TPOT (Table: 2), accompanied by a performance degradation of less than 20% in COMET scores. Concurrently, we observe close to 15% and 25% degradation in BLEU-1 and BLEU-2 respectively (see Table: 1).
2. Our results for English-to-Chinese translation demonstrate competitive performance compared to the Unified Skipping baseline - at $15\%$ skipping, the evaluation metrics on BLEU-1, BLEU-2 and COMET scores for our FiRST framework are equivalent to the Unified Skipping baseline.

For more detailed results of the machine translation task, comprising of all four BLEU scores (BLEU-1, BLEU-2, BLEU-3, BLEU-4), please refer to the Appendix A.4.

**CNN/DailyMail Dataset:**

1. There is an improvement in ROUGE-1, ROUGE-L scores (Table: 3) over LoRA fine-tuned base model suggesting that strategically skipping certain layers may even led to improved model performance. A 12% improvement in TPOT is observed for roughly 15% skipping. (Table: 2)

2. Our method outperforms Unified Skipping approach with more than 30% improvement in ROUGE-1 and ROUGE-L scores at 15% skipping rate.

Our experiments conclude the optimal skipping rate is around 15%, which maintains more than 80% of the original model performance (for WMT English-to-German and CNN/DM) and 75% of the original model performance for WMT English-to-Chinese - such a skipping level yields approximately 10-12% improvement in TPOT.

| Model Type | | ~Skipping (%) | English-to-German | | | English-to-Chinese | | |
|---|---|---|---|---|---|---|---|---|
| | | | BLEU-1 | BLEU-2 | COMET | BLEU-1 | BLEU-2 | COMET |
| Original Model (no skip) | Base + LoRA | 0 | 41.78 | 21.74 | 93.00 | 56.94 | 35.56 | 82.66 |
| | Base | 0 | 37.17 | 18.57 | 87.13 | 38.02 | 22.46 | 68.95 |
| Unified Layer Skipping | R + L | 15 | 28.92 | 10.64 | 59.34 | 46.61 | 25.01 | 69.58 |
| | R | 15 | 23.24 | 7.85 | 59.26 | 27.28 | 13.35 | 54.57 |
| | R + L | 25 | 15.67 | 3.36 | 31.69 | 34.90 | 15.75 | 50.59 |
| | R | 25 | 12.58 | 2.65 | 32.15 | 17.74 | 7.35 | 38.74 |
| | R + L | 35 | 6.44 | 0.77 | 22.05 | 7.51 | 2.10 | 20.25 |
| | R | 35 | 3.92 | 0.51 | 22.88 | 3.87 | 1.05 | 21.24 |
| Our Solution (FiRST) | R + L | 15 | 38.01 | 17.89 | 82.14 | 48.35 | 26.57 | 68.63 |
| | R | 15 | 28.83 | 11.80 | 67.74 | 17.55 | 8.68 | 42.76 |
| | R + L | 25 | 17.84 | 4.14 | 34.95 | 35.79 | 15.66 | 56.92 |
| | R | 25 | 9.67 | 1.37 | 26.01 | 11.01 | 3.23 | 25.45 |
| | R + L | 35 | 6.39 | 0.42 | 19.96 | 15.66 | 3.95 | 26.8 |
| | R | 35 | 3.70 | 0.14 | 21.41 | 6.13 | 1.54 | 22.89 |

Table 1: Quality Analysis on WMT (English-to-German and English-to-Chinese): BLEU-1, BLEU-2 and COMET scores for varying skipping % have been reported. Here, R + L corresponds to Router Augmentation followed by LoRA fine-tuning and R corresponds to router only (in the proposed FiRST framework). FiRST with finetuning, improves upon the input-agnostic baseline of Unified Skipping for all skipping levels - the improvement is more pronounced for English-to-German.

| Model Type | ~ Skipping (%) | English-to-German TPOT | English-to-Chinese TPOT |
|---|---|---|---|
| Base + LoRA | 0 | 1x | 1x |
| R + L | 15 | 0.90x | 0.88x |
| R + L | 25 | 0.82x | 0.83x |
| R + L | 35 | 0.69x | 0.68x |

| Model Type | ~Skipping (%) | CNN/DM TPOT |
|---|---|---|
| Base + LoRA | 0 | 1x |
| R + L | 15 | 0.88x |
| R + L | 20 | 0.81x |
| R + L | 27 | 0.76x |

Table 2: TPOT variation on WMT (left) and CNN/DM (right) for FiRST. These values are relative to LoRA fine-tuned base model. Fine-tuning is able to improve both TPOT and quality significantly.

| Model Type | | ~Skipping (%) | BERT F1 | Rouge-1 | Rouge-L |
|---|---|---|---|---|---|
| Original Model (no skip) | Base + LoRA | 0 | 84.87 | 28.46 | 16.99 |
| | Base | | 82.29 | 23.49 | 14.66 |
| Unified Layer Skipping | R + L | 15 | 84.25 | 24.35 | 14.30 |
| | R | | 80.30 | 16.61 | 10.95 |
| | R + L | 20 | 82.93 | 22.30 | 13.37 |
| | R | | 80.32 | 16.51 | 11.15 |
| | R + L | 27 | 80.28 | 15.94 | 9.89 |
| | R | | 77.43 | 10.97 | 7.68 |
| Our Solution (FiRST) | R + L | 15 | 85.14 | 31.80 | 20.13 |
| | R | | 81.25 | 20.20 | 13.01 |
| | R + L | 20 | 82.80 | 27.65 | 17.84 |
| | R | | 79.32 | 16.28 | 10.85 |
| | R + L | 27 | 77.50 | 14.65 | 10.45 |
| | R | | 75.60 | 9.39 | 6.92 |

Table 3: Quality Analysis on Summarization (CNN/DM dataset): BERT F1, Rouge-1 and Rouge-L scores are reported for varying skipping levels. FiRST with fine-tuning, improves upon Unified Skipping for all skipping levels on both Rouge-1 and Rouge-L and is competitive on BERT F1.

## 7 CONCLUSION

We provide a new algorithm and framework FIRST for layer selection corresponding to input sequence and task towards reducing latency in a quality aware manner. This operates in a KV cache compatible manner and handles batches of sequences, which are drawbacks in many existing work on early exit. We show significant reduction in latency with low degradation of quality on multiple tasks on well known open source datasets and demonstrate superior quality and latency over input agnostic baselines. In the future, we would like to extend our method to 1) improve the optimality of selection of layers, and 2) learn dynamically as new input sequences and tasks arrive.

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

## A APPENDIX

### A.1 TRAINING AND TESTING SPLIT

| | WMT | | Summarization |
|---|---|---|---|
| | **English-to-German** | **English-to-Chinese** | **CNN-DM** |
| **Train** | 3505 | 8983 | 3400 |
| **Validation** | 876 | 998 | 600 |
| **Test** | 2038 | 2038 | 11490 |

Table 4: Train-Validation-Test split for WMT and CNN datasets

### A.2 PROMPT DETAILS

The prompt structures used for both training and inference are as follows:

- For the machine translation task (English-to-German or English-to-Chinese), the following general prompt structure is used to train the routers and during final inference:

```
### Instruction:
Translate the following sentences from English to German.

### Input:
{Text to be translated}

### Response:
```

- For the summarization task (used in CNN/DailyMail dataset), the following prompt structure is utilized:

```
### Instruction:
Summarize the news article in around 100-200 words.

### Input:
{Article to be summarized}

### Response:
```

During the training of the LoRA module, task-aware training is applied. The expected translation or summary is appended after the ### Response section, making the model predict the response tokens following the "Response:\n".

### A.3 LAYER-WISE SKIPPING STATISTICS

Tables 5, 6, and 7 indicate the fraction of sequences that skip a particular block during the task. If the corresponding cell in a row shows a value of 0.8, it implies that 80% of the sequences skip this block. It is important to note that the decision regarding which block to skip varies across different datasets and tasks. Additionally, partial skipping in some blocks, with varying percentages, suggests that while some sequences consider the layer important, others do not and therefore skip it during the decoding phase.

Figures 4, 5, and 6 illustrate how various blocks are skipped when the model is adjusted to skip approximately 15% of the layers. These plots highlight which blocks are skipped more frequently, depending on the specific task and dataset being used.

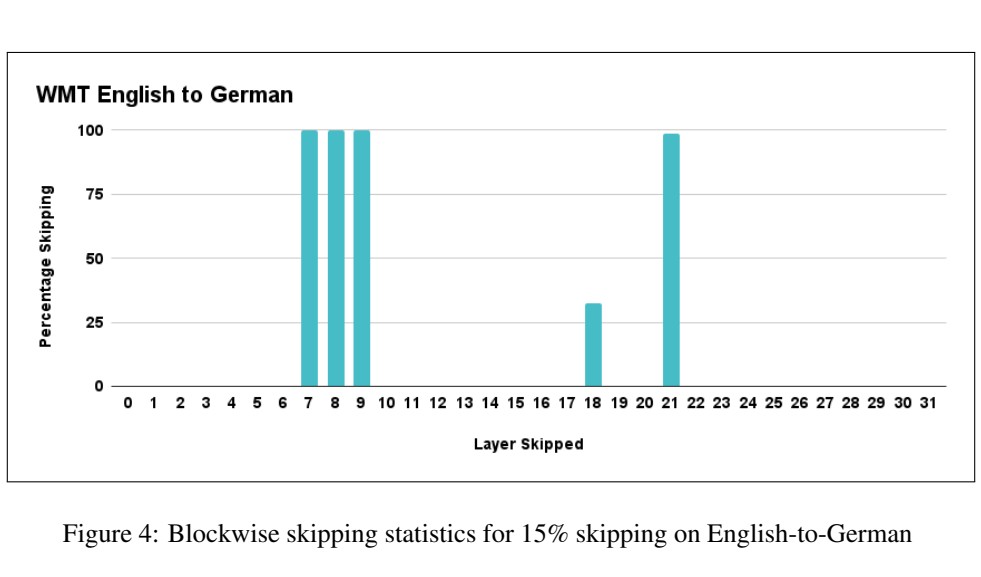

Figure 4: Blockwise skipping statistics for 15% skipping on English-to-German

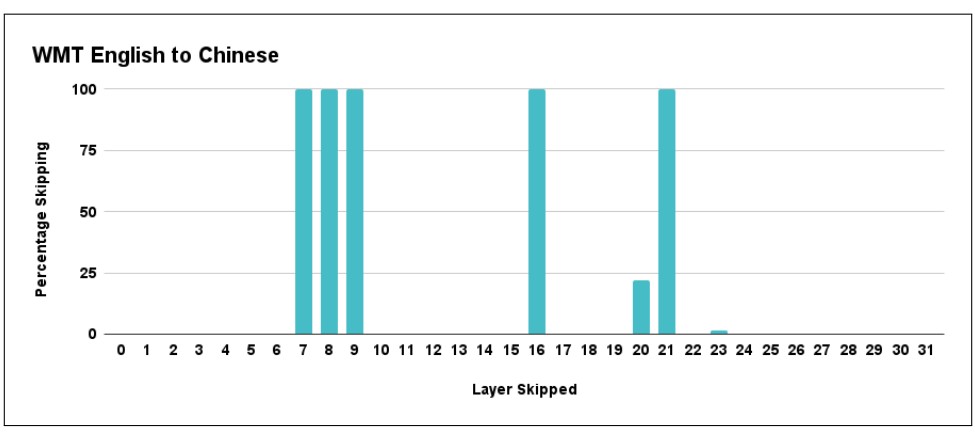

Figure 5: Blockwise skipping statistics for 15% skipping on English-to-Chinese

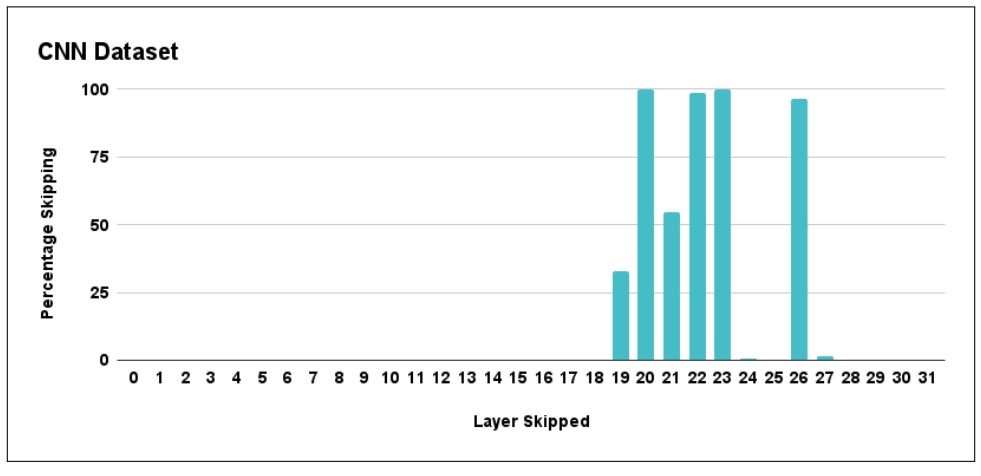

Figure 6: Blockwise skipping statistics for 15% skipping on CNN/DM

| Layer ↓ $\alpha \rightarrow$ | R 0.005 | R+L 0.005 | R 0.01 | R+L 0.01 | R 0.025 | R+L 0.025 |
|---|---|---|---|---|---|---|
| 0 | 0 | 0 | 0 | 0 | 0 | 0 |
| 1 | 0 | 0 | 0 | 0 | 0 | 0 |
| 2 | 0 | 0 | 0 | 0 | 0 | 0 |
| 3 | 0 | 0 | 0 | 0 | 0 | 0 |
| 4 | 0 | 0 | 0 | 0 | 0 | 0 |
| 5 | 0 | 0 | 0 | 0 | 0 | 0 |
| 6 | 0 | 0 | 0 | 0 | 0 | 0 |
| 7 | 1 | 1 | 1 | 1 | 1 | 1 |
| 8 | 1 | 1 | 1 | 1 | 1 | 1 |
| 9 | 1 | 1 | 0 | 0 | 0.0010 | 0.0132 |
| 10 | 0 | 0 | 0 | 0 | 0.8925 | 0.7267 |
| 11 | 0 | 0 | 0 | 0 | 0 | 0 |
| 12 | 0 | 0 | 1 | 1 | 1 | 1 |
| 13 | 0 | 0 | 0 | 0 | 0 | 0 |
| 14 | 0 | 0 | 0 | 0.0059 | 0 | 0.0025 |
| 15 | 0 | 0 | 1 | 0.9995 | 1 | 0.9936 |
| 16 | 0 | 0 | 0.0010 | 0.0245 | 1 | 0.9995 |
| 17 | 0 | 0 | 0 | 0 | 0 | 0 |
| 18 | 0.9779 | 0.3224 | 1 | 1 | 1 | 1 |
| 19 | 0 | 0 | 0.9985 | 0.9117 | 1 | 0.9961 |
| 20 | 0 | 0 | 1 | 1 | 1 | 0.9946 |
| 21 | 0.9985 | 0.9872 | 1 | 1 | 1 | 1 |
| 22 | 0 | 0 | 0 | 0 | 0 | 0 |
| 23 | 0 | 0 | 0.2414 | 0.0079 | 0.9975 | 0.9166 |
| 24 | 0 | 0 | 0 | 0 | 0 | 0 |
| 25 | 0 | 0 | 0 | 0 | 0 | 0 |
| 26 | 0 | 0 | 0.5731 | 0.0245 | 1 | 0.8602 |
| 27 | 0 | 0 | 0 | 0 | 0 | 0 |
| 28 | 0 | 0 | 0 | 0 | 0 | 0.0005 |
| 29 | 0 | 0 | 0 | 0 | 0 | 0 |
| 30 | 0 | 0 | 0 | 0 | 0 | 0 |
| 31 | 0 | 0 | 0 | 0 | 0 | 0 |
| Average Skipping | 0.1555 | 0.1347 | 0.2754 | 0.2492 | 0.3716 | 0.3595 |

Table 5: English-to-German: Skipping variation with Non-skip Penalization Loss coefficient $\alpha$

| Layer ↓ $\alpha \rightarrow$ | R 0.01 | R+L 0.01 | R 0.015 | R+L 0.015 | R 0.02 | R+L 0.02 |
|---|---|---|---|---|---|---|
| 0 | 0 | 0 | 0 | 0 | 0 | 0 |
| 1 | 0 | 0 | 0 | 0 | 0 | 0 |
| 2 | 0 | 0 | 0 | 0 | 0 | 0 |
| 3 | 0 | 0 | 0 | 0 | 0 | 0 |
| 4 | 0 | 0 | 0 | 0 | 0 | 0 |
| 5 | 0 | 0 | 0 | 0 | 0 | 0 |
| 6 | 0 | 0 | 0 | 0 | 0 | 0 |
| 7 | 1 | 1 | 1 | 1 | 1 | 1 |
| 8 | 1 | 1 | 1 | 1 | 1 | 1 |
| 9 | 1 | 1 | 1 | 1 | 1 | 1 |
| 10 | 0 | 0 | 0 | 0 | 0 | 0 |
| 11 | 0 | 0 | 0 | 0 | 0 | 0 |
| 12 | 0 | 0 | 0 | 0 | 1 | 1 |
| 13 | 0 | 0 | 0 | 0 | 0 | 0 |
| 14 | 0 | 0 | 0 | 0 | 0 | 0 |
| 15 | 0 | 0 | 0.4882 | 0.2812 | 0.1300 | 0.0015 |
| 16 | 1 | 1 | 1 | 1 | 1 | 1 |
| 17 | 0 | 0 | 0.0005 | 0 | 1 | 0.9990 |
| 18 | 0 | 0 | 0.7478 | 0.5584 | 1 | 1 |
| 19 | 0 | 0 | 0 | 0 | 0.9971 | 0.8690 |
| 20 | 0.2640 | 0.2184 | 1 | 1 | 1 | 1 |
| 21 | 1 | 0.9975 | 1 | 1 | 1 | 1 |
| 22 | 0 | 0 | 0 | 0 | 0.0005 | 0.0255 |
| 23 | 0.0029 | 0.0172 | 0.9549 | 0.9833 | 0.9975 | 0.9539 |
| 24 | 0 | 0 | 0 | 0 | 0 | 0 |
| 25 | 0 | 0 | 0 | 0 | 0 | 0 |
| 26 | 0 | 0 | 0.6830 | 0.0029 | 1 | 0.9190 |
| 27 | 0 | 0 | 0 | 0 | 0 | 0 |
| 28 | 0 | 0 | 0 | 0 | 0.5226 | 0.0015 |
| 29 | 0 | 0 | 0 | 0 | 0 | 0 |
| 30 | 0 | 0 | 0 | 0 | 0 | 0 |
| 31 | 0 | 0 | 0 | 0 | 0 | 0 |
| Average Skipping | 0.1646 | 0.1635 | 0.2773 | 0.2446 | 0.3952 | 0.3678 |

Table 6: English-to-Chinese: Skipping variation with Non-skip Penalization Loss coefficient $\alpha$

| Layer ↓ α → | R 0.03 | R+L 0.03 | R 0.035 | R+L 0.035 | R 0.04 | R+L 0.04 |
|---|---|---|---|---|---|---|
| 0 | 0 | 0 | 0 | 0 | 0 | 0 |
| 1 | 0 | 0 | 0 | 0 | 0 | 0 |
| 2 | 0 | 0 | 0 | 0 | 0 | 0 |
| 3 | 0 | 0 | 0 | 0 | 0 | 0 |
| 4 | 0 | 0 | 0 | 0 | 0 | 0 |
| 5 | 0 | 0 | 0 | 0 | 0 | 0 |
| 6 | 0 | 0 | 0 | 0 | 0 | 0 |
| 7 | 0 | 0 | 0 | 0 | 0 | 0 |
| 8 | 0 | 0 | 0 | 0 | 0 | 0 |
| 9 | 0 | 0 | 0 | 0 | 0 | 0 |
| 10 | 0 | 0 | 0 | 0 | 0 | 0 |
| 11 | 0 | 0 | 0 | 0 | 0 | 0 |
| 12 | 0 | 0 | 0 | 0 | 0 | 0 |
| 13 | 0 | 0 | 0 | 0 | 0.4359 | 0.5547 |
| 14 | 0 | 0 | 0 | 0 | 0 | 0 |
| 15 | 0 | 0 | 0 | 0 | 0 | 0 |
| 16 | 0 | 0 | 0 | 0 | 0 | 0 |
| 17 | 0 | 0 | 0 | 0 | 0 | 0 |
| 18 | 0 | 0 | 0.1889 | 0.1284 | 0.9997 | 0.9996 |
| 19 | 0.4282 | 0.3282 | 0.9991 | 0.9920 | 1 | 1 |
| 20 | 0.9986 | 0.9984 | 1 | 1 | 1 | 1 |
| 21 | 0.6825 | 0.5485 | 1 | 1 | 1 | 1 |
| 22 | 0.9936 | 0.9867 | 1 | 1 | 1 | 1 |
| 23 | 1 | 1 | 1 | 1 | 1 | 1 |
| 24 | 0.0011 | 0.0044 | 0.3891 | 0.4346 | 0.9265 | 0.9225 |
| 25 | 0 | 0 | 0 | 0 | 0.0016 | 0.0053 |
| 26 | 0.9712 | 0.9638 | 0.9997 | 0.9995 | 1 | 1 |
| 27 | 0.0179 | 0.0138 | 0.1463 | 0.1427 | 0.3742 | 0.3577 |
| 28 | 0 | 0 | 0 | 0 | 0.0089 | 0.0105 |
| 29 | 0 | 0 | 0 | 0 | 0 | 0 |
| 30 | 0 | 0 | 0 | 0 | 0 | 0 |
| 31 | 0 | 0 | 0 | 0 | 0 | 0 |
| Average Skipping | 0.1592 | 0.1514 | 0.2101 | 0.2093 | 0.2733 | 0.2766 |

Table 7: CNN/DM: Skipping variation with Non-skip Penalization Loss coefficient $\alpha$

## A.4 Detailed Result Table

Tables 8 and 9 present the detailed results of the machine translation task, reporting scores on all four BLEU metrics (BLEU-1, BLEU-2, BLEU-3, BLEU-4) and COMET. These tables highlight the performance across skipping percentages.

Table 10 indicates the improvement in average time to generate output tokens, specifically for the TPOT on GPU under both the router-only and LoRA+router configurations. Note that the latency improvement is significantly better in the LoRA+router case compared to the router-only case. Since the router model is not fine-tuned for the specific task, the number of tokens generated may vary.

Given that our solution applies skipping only during decoding, observing token generation during this phase is essential to evaluate TPOT improvements. Once LoRA is fine-tuned, the model generates more appropriate responses, resulting in a visible enhancement in TPOT.

| Model Type | | ~Skipping (%) | BLEU-1 | BLEU-2 | BLEU-3 | BLEU-4 | COMET |
|---|---|---|---|---|---|---|---|
| Original Model (no skip) | Base + LoRA | 0 | 56.94 | 35.56 | 23.19 | 16.02 | 82.66 |
| | Base | 0 | 38.02 | 22.46 | 13.85 | 9.14 | 68.95 |
| Unified Layer Skipping | R+L | 15 | 46.61 | 25.01 | 14.33 | 8.99 | 69.58 |
| | R | 15 | 27.28 | 13.35 | 7.08 | 4.25 | 54.57 |
| | R+L | 25 | 34.90 | 15.75 | 7.70 | 4.46 | 50.59 |
| | R | 25 | 17.74 | 7.35 | 3.52 | 2.06 | 38.74 |
| | R+L | 35 | 7.51 | 2.10 | 0.74 | 0.37 | 20.25 |
| | R | 35 | 3.87 | 1.06 | 0.37 | 0.20 | 21.24 |
| Our Solution (FiRST) | R+L | 15 | 48.35 | 26.57 | 15.80 | 10.27 | 68.63 |
| | R | 15 | 17.55 | 8.68 | 4.70 | 2.83 | 42.76 |
| | R+L | 25 | 35.79 | 15.66 | 7.99 | 4.77 | 56.92 |
| | R | 25 | 11.01 | 3.23 | 1.15 | 0.58 | 25.45 |
| | R+L | 35 | 15.66 | 3.95 | 1.43 | 0.75 | 26.80 |
| | R | 35 | 6.13 | 1.54 | 0.42 | 0.20 | 22.89 |

Table 8: English-to-Chinese: BLEU and COMET scores for varying skipping %

| Model Type | | ~Skipping (%) | BLEU-1 | BLEU-2 | BLEU-3 | BLEU-4 | COMET |
|---|---|---|---|---|---|---|---|
| Original Model (no skip) | Base + LoRA | 0 | 41.78 | 21.74 | 12.30 | 6.93 | 93.00 |
| | Base | 0 | 37.17 | 18.57 | 10.09 | 5.71 | 87.13 |
| Unified Layer Skipping | R+L | 15 | 28.92 | 10.64 | 4.60 | 1.95 | 59.34 |
| | R | 15 | 23.24 | 7.85 | 3.25 | 1.39 | 59.26 |
| | R+L | 25 | 15.67 | 3.36 | 1.01 | 0.33 | 31.69 |
| | R | 25 | 12.58 | 2.65 | 0.85 | 0.23 | 32.15 |
| | R+L | 35 | 6.44 | 0.77 | 0.12 | 0.02 | 22.05 |
| | R | 35 | 3.92 | 0.51 | 0.07 | 0.01 | 22.88 |
| Our Solution (FiRST) | R+L | 15 | 38.01 | 17.89 | 9.18 | 4.78 | 82.14 |
| | R | 15 | 28.83 | 11.80 | 5.66 | 2.93 | 67.74 |
| | R+L | 25 | 17.84 | 4.14 | 1.35 | 0.36 | 34.95 |
| | R | 25 | 9.67 | 1.37 | 0.33 | 0.05 | 26.01 |
| | R+L | 35 | 6.39 | 0.42 | 0.07 | 0.01 | 19.96 |
| | R | 35 | 3.70 | 0.14 | 0.01 | 0.00 | 21.41 |

Table 9: English-to-German: BLEU and COMET scores for varying skipping %

| Model Type | ~ Skipping (%) | TPOT GPU |
|---|---|---|
| Base + LoRA | 0 | 1x |
| R+L | 15 | 0.90x |
| R | 15 | 1.04x |
| R+L | 25 | 0.82x |
| R | 25 | 0.92x |
| R+L | 35 | 0.69x |
| R | 35 | 0.77x |

| Model Type | ~Skipping (%) | TPOT GPU |
|---|---|---|
| Base + LoRA | 0 | 1x |
| R+L | 15 | 0.88x |
| R | 15 | 0.98x |
| R+L | 25 | 0.78x |
| R | 25 | 0.89x |
| R+L | 35 | 0.68x |
| R | 35 | 0.80x |

Table 10: English-to-German (left) and English-to-Chinese (right) TPOT variation

