# OpenReview forum: "FiRST: Finetuning Router-Selective Transformers for Input-Adaptive Latency Reduction"
_ICLR.cc/2025/Conference — ICLR 2025 Conference Withdrawn Submission_

### Official Review · Reviewer_D1SP · 2024-10-31

**Soundness:** 2
**Presentation:** 2
**Contribution:** 2
**Rating:** 3
**Confidence:** 5

**Summary:**

The paper propose a pipeline for both training and inference named FIRST. The proposed framework includes layer-specific routers which are input-dependent selectors that can decide which layers are worth for inference. The proposed framework is compatible to the KV cache and batch decoding, which means there is no additional computation needed for handling both processes.
The paper also import LoRA method for further improvement of the model performance and selection strategy.
The experimental results on Machine Translation and Summarization tasks show the effectiveness of the proposed method.

**Strengths:**

1. This paper proposes the router for the layer selection, which allows the fine-tuning to optimize the selection strategy.
2. The LoRA is adopted for further optimization, which shows helpful for the selection strategy optimization.

**Weaknesses:**

1. The paper novelty is limited. The layerly redundancy has been explored by a lot of works [1] [2] [3], while none of those works were introduced for comparison in the results tables.
2. The proposed routers require additional parameters and corresponding training process, which is not kind of efficient and effective compared to those works [1] [2] [3] that do not require additional parameters and training. Meanwhile, the router selection has been adopted for DiT models in work [4].
3. The method proposed by the paper is kind of general method, while the paper only focuses on the Machine Translation and Summarization tasks, which hurts the motivation of this method.
4. According to the Table 1, Table 2, Table 3, the LoRA method seems to have strong improvement for the proposed method, while the ablation study of LoRA only results is absence, which may cause the misunderstanding that the effectiveness is brought by the LoRA rather than the proposed method.


[1] Compressing Large Language Models by Streamlining the Unimportant Layer \
[2] Shortened LLaMA: Depth Pruning for Large Language Models with Comparison of Retraining Methods \
[3] Sheared LLaMA: Accelerating Language Model Pre-training via Structured Pruning \
[4] Learning-to-Cache: Accelerating Diffusion Transformer via Layer Caching

**Questions:**

1. How about the model performance on other normal tasks and datasets including WikiText2, C4, and other common sense reasoning datasets including BoolQ, PIQA, HellaSwag, WinoGrande, ARC-easy, ARC-challenge, and OpenbookQA?
2. How about the model performance of other layer pruning LLM works [1] [2] [3] ?
3. What is the motivation of adopting this framework for Translation and Summarization tasks?



[1] Compressing Large Language Models by Streamlining the Unimportant Layer \
[2] Shortened LLaMA: Depth Pruning for Large Language Models with Comparison of Retraining Methods \
[3] Sheared LLaMA: Accelerating Language Model Pre-training via Structured Pruning

---

### Official Review · Reviewer_16Gx · 2024-11-01

**Soundness:** 2
**Presentation:** 3
**Contribution:** 2
**Rating:** 3
**Confidence:** 4

**Summary:**

The paper presents a method called FIRST (Finetuning Router-Selective Transformers) for reducing inference latency in autoregressive LLMs by adaptively selecting transformer layers to process during inference. This approach utilizes layer-specific routers that decide in real-time which layers to skip based on the input sequence, aiming to maintain model performance while reducing computational overhead. The method is enhanced with LoRA adapters for fine-tuning on task-specific data, purportedly improving performance while preserving latency benefits. Eperiments are reported to validate the efficacy of FIRST across different tasks, such as machine translation and summarization.

**Strengths:**

1. The idea of using routers to dynamically skip layers based on input characteristics can help address the latency issues prevalent in deploying LLMs on resource-constrained devices.

2. The use of LoRA adapters for fine-tuning while layer skipping helps mitigate the potential degradation in model performance.

**Weaknesses:**

1. The method is only experimented with on the Llama-3-8B model and classical Machine Translation and Summarization tasks, omitting newer, more challenging benchmarks such as commonsense reasoning, MMLU, and BIG-bench hard, which are critical for evaluating the generality of LLMs.

2. Given that current popular LLM benchmarks often contain many subtasks, the proposed method's adaptability and generalization across such varied tasks might be limited.

3. The necessity for a two-step training process for each task could be time-consuming. Methods that support post-training or zero-shot capabilities might be more desirable for LLM applications.

4. The paper lacks a detailed discussion on the complexity and computational overhead of training these routers, which is crucial for assessing the practicality of the FIRST approach.

**Questions:**

1. Could the authors test the method on other LLM benchmarks, such as commonsense reasoning or MMLU, to assess its effectiveness on a broader range of tasks?

2. How does the router handle dependencies between layers, especially when earlier layers are skipped, which might be crucial for the computations of subsequent layers?

3. How does introducing probabilistic layer skipping affect the gradient flow and stability of training?

---

### Official Review · Reviewer_AZpz · 2024-11-04

**Soundness:** 2
**Presentation:** 3
**Contribution:** 2
**Rating:** 3
**Confidence:** 4

**Summary:**

This paper introduces FiRST, a framework to reduce inference latency in Large-Language Models (LLMs) by selectively skipping specific layers for each input sequence. FiRST uses layer-specific routers to determine which layers to skip during the prefill stage and skip corresponding layers during the decode stage to accelerate inference. A LoRA adapter has also been introduced to further boost the performance for task-specific fine-tuning. Experimental results show that FiRST can reduce latency while maintaining competitive performance.

**Strengths:**

1. The paper adopts the idea of the router skipping specific layers for efficient LLM inference while maintaining performance.
2. The paper adopts the LoRA module to maintain the model's performance on different tasks.
3. The paper is well-written and easy to follow.

**Weaknesses:**

1. Experiments were only conducted on fine-tuned datasets. There was no zero-shot performance shown for the proposed method, for example, PPL and accuracy on the `lm-evaluation-harness` benchmark.
2. Lack of baseline, the FiRST only compared with Unified Layer Skipping. Beyond layer skipping, there are many methods targeting LLMs inference efficiency, e.g., [1][2][3]
3. The overhead of the proposed is not discussed.

[1] SLEB: Streamlining LLMs through Redundancy Verification and Elimination of Transformer Blocks. Song, Jiwon and Oh, Kyungseok, et al., ICML 2024

[2] QUEST: Query-Aware Sparsity for Efficient Long-Context LLM Inference. Jiaming Tang and Yilong Zhao and et al., ICML 2024

[3] SliceGPT: Compress large language models by deleting rows and columns. Saleh Ashkboos, Maximilian L. Croci, et al., ICLR 2024

**Questions:**

1. How does the method determine the number of layers to skip? Is it determined purely by the router, or can the user define it?
2. How does the speed-up compare to the unified layer skipping?
3. What is the overhead of the proposed method since it introduces additional modules? It should be easy to obtain by applying random layer skip without layer.

---

### Official Review · Reviewer_m71U · 2024-11-08

**Soundness:** 2
**Presentation:** 2
**Contribution:** 3
**Rating:** 3
**Confidence:** 5

**Summary:**

This paper provides an algorithm FIRST that reduces inference latency using layer selection corresponding to input sequences and tasks.
The authors evaluate FIRST on two tasks, including machine translation and summarization.

**Strengths:**

- The writing is clear
- FIRST is evaluated on two different tasks including machine translation and summarization.

**Weaknesses:**

- Ablation study is missing.
- Evaluation is insufficient. (1) It will be better if more than one dataset is evaluated for each task (machine translation and summarization). (2) Besides the Llama-3-8B base model, more model architectures should be evaluated. (3) The baselines are the base model (Llama-3-8B) with and without LoRA fine-tuning, which is not sufficient. In addition, besides unified skipping, more related methods should be compared with the proposed method.
- Figures should be plotted better. The font in Figures 2 and 3 is small.

**Questions:**

- In the related work section, how does the proposed method relate to the quantization-aware training? They both aim to reduce mode latency. However, quantization-aware training reduces bit-widths with retraining, and the proposed method may not be in that research area.
- How do the values of $\lambda$ and $\alpha$ of the total loss affect the performance? Why is $\lambda$ fixed at 0.01? In the training settings, the discussion about the settings of $\alpha$ is missing.  Are lora_alpha and $\alpha$ the same thing?

---

### Note · Authors · 2024-11-25

I have read and agree with the venue's withdrawal policy on behalf of myself and my co-authors.